

# Assessing the impact of different liquid water permittivity models on the fit between model and observations

Katrin Lonitz[1] and Alan J. Geer[1]

[1]European Centre for Medium-Weather Range Forecasts, Shinfield Park, RG2 9AX Reading, United Kingdom

*Correspondence to:* Katrin Lonitz (katrin.lonitz@ecmwf.int)

**Abstract.** Permittivity models for microwave frequencies of liquid water below 0°C (supercooled liquid water) are poorly constrained due to limited laboratory experiments and observations, especially for high microwave frequencies. This uncertainty translates directly into errors in retrieved liquid water paths of up to 80%. This study investigates the effect of different liquid water permittivity models on simulated brightness temperatures by using the all-sky assimilation framework of the Integrated

Forecast System. Here, a model configuration with an improved representation of supercooled liquid water has been used. The comparison of five different permittivity models with the current one shows a small mean reduction in simulated brightness temperatures of at most 0.15 K at 92 GHz on a global monthly scale. During austral winter differences occur more prominently in the storm tracks of the southern hemisphere and in the Intertropical Convergence Zone with values of around 0.5 K to 1.5 K. For most permittivity models the fit to observations is slightly improved compared to the default one. The permittivity model by

Rosenkranz (2015) is recommended to be used inside the observation operator RTTOV-SCATT for frequencies below 183 GHz.

## 1  Introduction

The occurrence of liquid water for temperatures below 0°C (supercooled liquid water) is typical for clouds in the higher latitudes (e.g. in frontal systems and cold-air outbreak regions). Inside clouds liquid water can exists down to -40°C (Heymsfield

et al., 1991). Due to a lack of laboratory experiments and observations the constraint on absorption properties of supercooled liquid water is poor. More precisely, the permittivity (or dielectric constant) of liquid water, which is one of the key factors determining the absorption in the microwave band, is poorly known for these low temperatures and, hence, existing liquid water permittivity models differ substantially. Recently, two new liquid water permittivity models by Rosenkranz (2015) and Turner et al. (2016) have been published. Both models are based partly on findings by Kneifel et al. (2014), who compared existing

permittivity models (e.g. Stogryn et al., 1995; Ellison, 2007) with new observations from ground-based microwave radiometers between 31 and 225 GHz for clouds from 0°C to -33°C. Kneifel et al. (2014) found that the different liquid water permittivity models agree fairly well with each other between 0°C and -15°C, but differ by 25% and more at lower temperatures (i.e. for supercooled liquid water), especially for frequencies higher than 35 GHz.





Liquid water permittivity models are usually compared with observations undertaken at certain locations or with laboratory results. In this study we quantify the global and local impact of the different permittivity models for pure liquid water in the context of the assimilation of microwave imager observations sensitive to clouds, humidity and precipitation using the Integrated Forecast system (IFS) of the ECMWF. Since 2009, ECMWF has used an all-sky framework for the assimilation of microwave radiances (Bauer et al., 2010), which means that these observations are assimilated under clear, cloudy and precipitating conditions.

To allow a thorough study about the impact of different liquid water permittivity models for the simulation of microwave imager observations the assimilation system and the forecast model have to have some special characteristics. First, the assimilation has to allow the simulation of observations under cloudy conditions, which is the case for microwave imager observations inside the IFS. Second, the forecast model should have skill in representing areas which are of most interest when it comes to studying the effect of absorption properties of liquid water. As shown by Kneifel et al. (2014) these are areas containing supercooled liquid water. A recent study by Forbes et al. (2016) showed, however, that one of the long-standing model biases in the shortwave radiation is related to a lack of supercooled liquid water in cold-air outbreak regions in the IFS. This bias is also well known for other numerical weather prediction (NWP) and climate models (e.g. Bodas-Salcedo et al., 2016). For this reason, a special model configuration of the IFS has been used, incorporating improvements which allow the generation of more supercooled liquid water (see section 2.3).

Accurate absorption properties of cloud liquid water are needed for the construction of a reliable observation operator for microwave observations. Uncertainties in e.g. absorption properties of cloud liquid water inside the observation operator have, therefore, the potential to introduce systematic situation dependent errors. For the all-sky assimilation of microwave radiances inside the IFS the observation operator RTTOV-SCATT (section 2.1) is used. It converts physical variables, e.g. humidity and temperature from the model into observed variables, e.g. brightness temperatures. At the moment the liquid water permittivity model by Liebe (1989) is used inside RTTOV-SCATT. However, as Kneifel et al. (2014) have shown newer permittivity models might be more suitable, especially in areas with supercooled liquid water (e.g cold-air outbreaks) where a high uncertainty among the different permittivity models exists.

There is a high need to conduct such a "closure" study about the best choice of permittivity model for liquid water. We examine this issue using a high-quality NWP model for the first time. This approach enables the quantification of the impact of the different permittivity models globally and the comparison of the effect with other independent observations. In detail, this closure study examines the effect of six different formulations of permittivity on simulated brightness temperatures and departures, especially for areas where clouds with supercooled liquid water prevail. First, the observation operator, the usage of data and the setup of this study are explained. Next, the impact on absorption and simulated brightness temperature are shown for the different permittivity models. Eventually, the best model based on monitoring and assimilation experiments is chosen.



## 2 Methodology

### 2.1 Observation Operator RTTOV-SCATT

RTTOV-SCATT is the observation operator for the microwave radiative transfer in cloudy, precipitating and clear skies (Bauer et al., 2006) and is a component of the RTTOV package (Radiative Transfer model for Television Infrared Observation Satel-
lite Operational Vertical sounder, Saunders et al., 1999). The radiative transfer equation is solved using the delta-Eddington approximation which produces mean errors of less than 0.5 K at the targeted microwave frequencies between 10 and 200 GHz (Bauer et al., 2006). The final all-sky brightness temperature is a weighted average of the brightness temperature from a cloudy and a clear sub-column, where the weighting is done using an effective cloud fraction (Geer et al., 2009).

    Generally speaking radiation in the atmosphere can be absorbed or scattered by atmospheric particles, like aerosols, atmo-
spheric gases and hydrometeors. Which one of these processes dominates depends on the frequency, the size and shape of the particles and in case of conducting materials like hydrometeors on the relative permittivity. We use Mie-theory to compute scattering and absorption properties of cloud liquid hydrometeors which are assumed to be homogeneous.

    In order to solve the radiative transfer equation the bulk optical properties of the atmosphere have to be known at each model level. Given optical properties of a single particle the bulk optical properties, i.e. extinction coefficient, scattering coefficient
and average asymmetry parameter can be computed by integration across a size distribution. This is done in form of look-up tables for different frequencies, temperatures and liquid/ice water contents for each hydrometeor type; in the IFS these are rain, snow, cloud water and cloud ice (for more information see: Bauer, 2001; Geer and Baordo, 2014). For cloud droplets scattering in the microwave regime is generally negligible and, hence, their extinction is equal to the absorption. However, for rain drops or snow Mie scattering occurs given that the ratio between their size and the wavelength can be much larger than for cloud
droplets.

    The absorption of liquid clouds depends amongst other things on the relative permittivity of water. Permittivity is a measure of the collective motion of the molecular dipole moments under the influence of an electric field, and consists of a real (scattering) component, and an imaginary (absorption) component. How strong the permittivity is depends on frequency, pressure, temperature (and salinity, which is 0 for pure water), as illustrated later in section 3. In this study, different permittivity formu-
lations of liquid hydrometeors (e.g. cloud droplets and rain drops) are examined. The permittivity formulation inside the ocean surface emissivity model FASTEM 6 (Kazumori and English, 2015) remains unchanged.

### 2.2 Specifications of microwave observations

To study the effect of the different liquid water permittivity models this study analyses mainly changes in simulated brightness temperature from SSMIS-F17 (Special Sensor Microwave Imager Sounder onboard the Defense Meteorological Satel-
lite Program satellite F17; DMSP-F17, Kunkee et al., 2008). As already mentioned, microwave imagers and microwave humidity sounders are assimilated under cloudy, precipitating and clear-sky (all-sky) conditions using the IFS. Currently, this includes instruments like, GMI (GPM Microwave Imager), AMSR2 (Advanced Microwave Scanning Radiometer 2), MHS (Microwave Humidity Sounding), SAPHIR (Sondeur Atmospherique du Profil d'Humidite Intertropicale par Radiometrie),



MWHS-2 (Micro-Wave Humidity Sounder-2) and of course SSMIS. Other microwave sensors, like AMSU-A (Advanced Microwave Sounding Unit - A) and ATMS (Advanced Technology Microwave Sounder) are still assimilated in clear-sky conditions. Alongside this, a suite of other data, e.g. radiance from hyperspectral Infrared Sounders, atmospheric motion vectors, radiosondes and aircraft data are assimilated.

In the all-sky system, microwave imager observations are only assimilated over ocean, whereas microwave humidity sounder observations at 183 GHz are assimilated over ocean and land. For frequencies $186 \pm 6$ GHz and below data is restricted to $60°$S and $60°$N and excludes ocean areas with sea ice. Higher peaking microwave humidity sounding channels are assimilated over ocean and land globally, also above sea ice. Areas with high orography are also excluded for microwave humidity sounder observations. Furthermore, microwave imager data is averaged (or "superobbed") to about 80 km x 80 km boxes in order to

match the effective resolution of cloudy and precipitating systems inside the forecast model. Additionally, microwave imager data is screened in some areas because of systematic model biases in e.g. cold-air outbreak regions (Lonitz and Geer, 2015). The data is also thinned to about 100 km. Further details of the all-sky microwave imager assimilation at ECMWF can be found in Bauer et al. (2010); Geer et al. (2017b).

     A specific observation error model was designed for the assimilation of microwave observation in all-sky conditions. Here,

the observation error is based on the 'symmetric' cloud amount (C37) which is an average of the observed and simulated cloud amount, represented in a cloud proxy variable that goes from 0 to 1. For SSMIS-F17 an observation error of 1.8 K is used in clear-sky conditions (C37<0.02), which increases linear up to 18 K for very cloudy situations with C37>0.42. The higher the observation error the less impact the observation has on the analysis. More details can be found in Geer and Bauer (2011).

### 2.3   Setup of Models and Experiments

#### 2.3.1   Liquid water permittivity models

Six different permittivity models incorporated into the observation operator RTTOV-SCATT (version 11.2) have been tested in the all-sky assimilation of microwave radiances. As stated above, only the formulation of permittivity of pure liquid water clouds and rain has been altered. Table 1 lists the acronyms for the different permittivity models as used in the remainder of this paper.

All liquid water permittivity models are based on laboratory data as well as field experiments when available. These observations have been used to construct a model using a multiple Debye formulation (Debye, 1929) to describe the different forms of motion of the molecular dipole moments, e.g. reorientation and bending also referred to as relaxation terms. The current permittivity model for liquid water Liebe (1989), along with Liebe et al. (1993), Stogryn et al. (1995) and Turner et al. (2016) utilise a double Debye formulation, whereas Rosenkranz (2015) and Ellison (2007) apply three relaxation terms to be able

to describe two modes of bending instead of just one. However, only Liebe (1989) and Liebe et al. (1993) are constructed explicitly for suspended water droplets, whereas the other models do not make special considerations or are constructed based on laboratory experiments with bulk water, as for example Ellison (2007).





Liebe (1989), Liebe et al. (1993) and Rosenkranz (2015) have been constructed to be valid up to 1 THz, whereas Stogryn et al. (1995) and Turner et al. (2016) are only valid up to 500 GHz. Ellison (2007) constructed a permittivity model to be valid up to 25 THz. Therefore, his permittivity model takes additional to the three relaxation terms two resonance terms into account due to stretching of intramolecular hydrogen bonds around 4 THz and librational motions of water molecules around 11 THz.

Most of the models claim validity also below 0°C (except Ellison, 2007), even though observations for supercooled liquid water are rare. Only the two most recent permittivity models for microwave frequencies Rosenkranz (2015) and Turner et al. (2016) incorporated a new observational dataset by Kneifel et al. (2014) measured at temperatures well below 0°C. Hence, Rosenkranz (2015) and Turner et al. (2016) are believed to be more accurate at temperatures below 0°C than earlier models from Liebe (1989), Liebe et al. (1993), Stogryn et al. (1995) and Ellison (2007). For more information about the basis and

settings of the different liquid water permittivity models the reader is advised to read through the literature listed in Tab. 1.

**Table 1.** List of different liquid water permittivity models and how they are referenced within this paper.

| permittivity model | reference |
| --- | --- |
| Liebe (1989) | Liebe89 |
| Liebe et al. (1993) | Liebe93 |
| Stogryn et al. (1995) | Stogryn95 |
| Ellison (2007) | Ellison07 |
| Rosenkranz (2015) | Rosenkranz15 |
| Turner et al. (2016) | TKC16 |

### 2.3.2    Forecast model

In order to evaluate the quality of the different liquid water permittivity models the simulated brightness temperatures are compared with the observed brightness temperatures from SSMIS-F17. Nevertheless, to make a fair comparison it is essential to use a suitable atmospheric model for which the liquid water in clouds and rain is realistically represented compared to the

real world.

Until IFS cycle 43R1, convective mixed-phase clouds have been represented by a fixed global diagnostic temperature-dependent function. That means for temperatures above 0°C cloud water was considered liquid, and below -23°C cloud water was considered ice. Between 0°C and -23°C there existed a decreasing proportion of liquid water and ice. In reality, however, a cloud can consist completely of (supercooled) liquid water below 0°C depending on the evolution of the cloud and its

environment. In IFS cycle 43R3 the lower threshold for the convective mixed phase was lowered to -38°C to meet findings by Heymsfield et al. (1991) with allowing additional detrainment of rain and snow (ECMWF, 2017). In the most current IFS cycle 45R1 the model physics have been altered to allow the generation of purely supercooled liquid water for surface driven shallow convection, whereas the mixed phase formulation still applies for deep and congestus clouds.





In this study, a model configuration using a modified version of the IFS cycle 43R3 with a horizontal resolution of approximately 16 km (T639 in spectral terms) and 137 vertical levels is used. This model configuration is based on IFS cycle 43R3 but utilises the 45R1 model physics, which allow the generation of more supercooled liquid water inside surface driven shallow convection clouds down to -38°C. This setup allows to study the sensitivity of the different liquid water permittivity

models for temperatures well below 0°C inside a NWP model, which would have not been possible before due to a lack of supercooled liquid water (Forbes et al., 2016). However, we know that not allowing the generation of purely supercooled liquid water congestus clouds or deep convection is one limitation of this formulation which has to be addressed in the future. This model configuration is used for all monitoring and assimilation experiments.

### 2.3.3 Experiments

The first set of experiments are monitoring experiments, which monitor a change in first guess departures without generating a new analysis and forecast. These experiments are used in sections 3 and 4. They enable the examination of the change in the simulated brightness temperature (or first guess [FG]) due to a change in the observation operator only and not through subsequent changes in the analysis field that would result from a full-cycling data assimilation system. All monitoring experiments use the same parent 43R3 experiment with 45R1 model physics assimilating additionally microwave imager data in cold-air

outbreak areas and in areas with a total water vapour content below $8\,\mathrm{kg\,m^{-2}}$ (these are normally screened, see Lonitz and Geer, 2015). Furthermore, to allow for a greater sample no thinning of the microwave imager data has been applied as done operationally. The experiments have run from 25 July to 31 August 2016 covering times where clouds with supercooled liquid water prevail in the mid- to high-latitudes of the southern hemisphere. The analysed time frame covers 1 to 31 August 2016.

The second set of experiments allows fully cycled data assimilation and is used to evaluate the impact of the choice in liquid

water permittivity model on forecast scores and fits to observations (section 5). All experiments use the same setup run using IFS cycle 43R3 with 45R1 model physics. Two assimilation experiments are carried out, one for which cold-air outbreak areas are screened (**screen**) and one for which data has also been assimilated in these regions (**plusSLW**). The experiments ran from 1 June to 30 September 2016. A summary of all experiment types is giving in Tab. 2.

**Table 2.** List of different experiment setups and their details, which have been run using different liquid water permittivity models.

| experiment type | microwave imager data usage different to operational configuration | name | analysed time |
|---|---|---|---|
| monitoring | additional microwave imager data in cold-air outbreak areas and in areas with low water vapour, no thinning of microwave imager data | | 1-31 Aug. 2016 |
| assimilation | - | **screen** | 1 Jun. - 30 Sep. 2016 |
| | additional microwave imager data in cold-air outbreak areas and in areas with low water vapour | **plusSLW** | 1 Jun. - 30 Sep. 2016 |





## 3 Changes in absorption and brightness temperatures

As mentioned in section 1 the largest changes from using different liquid water permittivity models are expected for high microwave frequencies (larger than 35 GHz) and in areas of supercooled liquid water clouds, as shown e.g. by Cadeddu and Turner (2011). Little impact is expected for precipitation with supercooled liquid water in these experiments. This is because

supercooled drizzle does not yet exist inside the model and supercooled rain drops exist only for very few situations just below 0°C (personal communication with Richard Forbes, ECMWF). Here, we investigate how the different permittivity formulations modulate absorption properties of liquid water and simulated brightness temperatures at different frequencies.

### 3.1 Absorption properties

Fig. 1 shows how absorption varies with temperature for a liquid-water cloud with 0.1 g/m$^3$ water content. As expected, the

largest variations in absorption occur for high microwave frequencies: 92 GHz and higher (Fig. 1). The higher the frequency the more the spread between the models can be seen for higher temperatures. Here, the largest spread can be seen for temperature below 0°C (273 K). Most of the models show slightly smaller values in absorption compared to Liebe89 with two exceptions in Ellison07 for temperatures between 255 K and 290 K and in Liebe93 for temperatures below 255 K, both for frequencies of 92 GHz and higher.

Fig. 2 shows variation with frequency for up to 1 THz. For temperatures around 0°C (Fig. 2b) the absorption increases with frequency for all permittivity formulations. The two most recent permittivity models Rosenkranz15 and TKC16 give about 50% of the absorption compared to Liebe89 for frequencies around the 183 GHz water vapour absorption line. Quite large differences can be seen for higher frequencies above 200 GHz. At 270 K all permittivity models show larger absorption values at 1 THz compared to 200 GHz with e.g. values twice as high for TKC16 and almost four times higher for Rosenkranz15 (Fig.

2b). At 240 K the discrepancy between the models is even higher (Fig. 2a). The absorption given by TKC16 seems to saturate for frequencies above 92 GHz, whereas for all the other permittivity models absorption increases with frequency throughout the whole frequency spectrum. Here, Rosenkranz15 shows the largest increase with frequency, having an absorption value of about 0.65 km$^{-1}$ at 1 THz (Fig. 2a). These main differences between Rosenkranz15 and TKC16 may be due to the subset of observations used to built the models, the differences in the Debye formulations or the method to fit the absorption model

coefficients. We think that the combination of a third relaxation term and fitting observations for frequencies up to 1 THz for Rosenkranz15 explains most of the differences in the higher frequency spectrum compared to TKC16, which only uses two relaxation terms and is constructed to be valid up to 500 GHz.

### 3.2 Effect of liquid water permittivity models on simulated brightness temperature

Results from the monitoring experiments show that a reduced absorption decreases the simulated brightness temperatures for

some frequencies. This can be seen in the mean difference in simulated brightness temperatures from various monitoring experiments using permittivity models at 37 GHz, v-polarised (37 v) , 92 GHz, v-polarised (92 v) and 150 GHz, h-polarised (150 h) co-located to SSMIS-F17 observations. Tab. 3 gives an overview of mean differences in the northern hemisphere, in







**Figure 1.** Absorption as a function of temperature for liquid water clouds with $0.1 \, \text{g/m}^3$ water content for different microwave frequencies.

the tropics and in the southern hemisphere. Most permittivity models show a small mean reduction in brightness temperature compared to Liebe89 in all regions; but especially in the southern hemisphere during austral winter. The largest (but still quite small) mean deviation from Liebe89 is found in the southern hemisphere for TKC16 at 92 v with a mean reduction of 0.288 K in simulated brightness temperature. The smallest difference is found for Liebe93 of about 0.003 K at 150 h in the southern
5 hemisphere.

Fig. 3, Fig. 4 and Fig. 5 show the geographical distribution of mean differences between the different permittivity models in simulated brightness temperature compared to Liebe89 at 37 v, 92 v, and 150 h, respectively. The largest differences occur predominately in the mid-latitudes and to a minor extent around the Intertropical Convergence Zone (ITCZ), which is linked





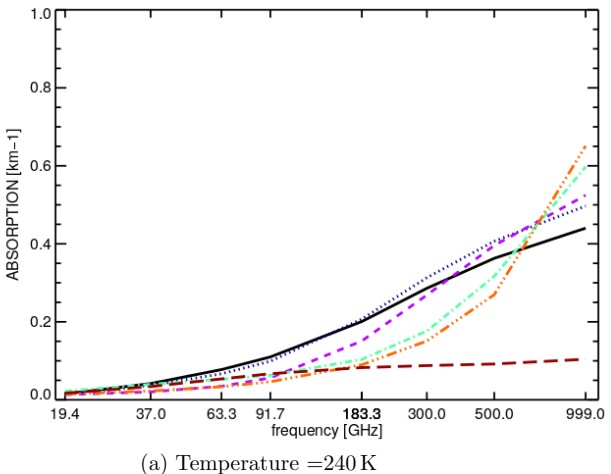
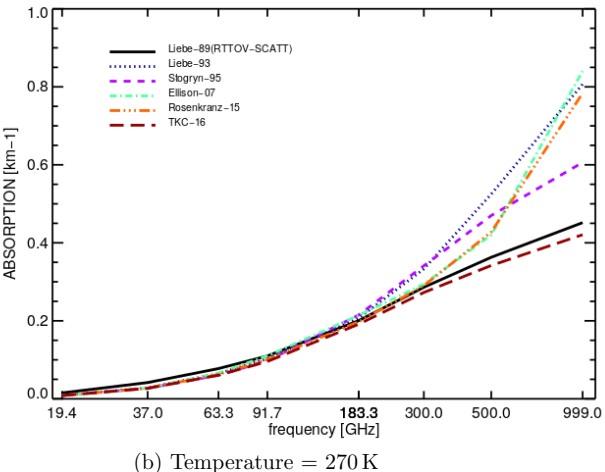

(a) Temperature = 240 K           (b) Temperature = 270 K

**Figure 2.** Absorption as a function of different microwave frequencies for liquid water clouds with 0.1 g/m$^3$ water content at temperatures of a) T=240 K and b) T=270 K. For the construction of these figures absorption values at 19.4 GHz, 37 GHz, 63.3 GHz, 91.7 GHz, 300 GHz, 500 GHz and 999 GHz have been computed. Values of absorption between these frequencies are linearly interpolated.

to the higher occurrence of supercooled liquid water in these regions. In the ITCZ deep reaching convective clouds prevail, which contain some supercooled liquid water. However, supercooled liquid water is clearly more frequent and influential on the simulated brightness temperatures at higher latitudes. Here, supercooled liquid water is found in fronts and in cold-air outbreak areas, which accounts for areas with largest changes (as shown in section 3.3). In the higher latitudes, Stogryn95,

Rosenkranz15 and TKC16 show a reduction in simulated brightness temperature at frequencies up to 150 GHz compared to Liebe89. Only Liebe93 shows an increase at 150 h despite a decrease at lower frequencies, and Ellison07 shows an increase at 92 v and 150 h despite a decrease at 37 v. This increase in brightness temperatures at high frequencies is due to higher absorption for temperatures below 260 K in case of Liebe93 and due to higher absorption for temperatures around 270 K in case of Ellison07 compared to Liebe89 (see Fig. 1c and Fig. 1d).

The sensitivity of absorption to the liquid water permittivity formulations is largest for high frequencies, as would be expected from Fig. 2. However, simulated brightness temperatures change only little for most regions at these frequencies, as can be seen for 183 ± 6 GHz, h-polarised (183 ± 6 h) in Fig. 6. The reason for this behaviour is based in the weighting function at 183 ± 6 h peaking around 700 hPa, which makes it less sensitive to lower lying supercooled liquid water clouds and more susceptible to the occurrence of snow or higher level clouds, which are predominately composed out of ice. In other words, the

radiative transfer effects at 183 GHz are dominated by scattering from frozen hydrometeors. At 183 ± 6 h the only large differences in simulated brightness temperature are found in a few areas along the ITCZ or the western Pacific Ocean associated with a higher occurrence of deep tropical convection, which must contain some supercooled liquid water.

     In general, larger mean differences between the permittivity models can be seen for frequencies up to 150 GHz. The larger differences in absorption at frequencies up to 150 h GHz goes in hand with the shift to higher temperatures in the spread among





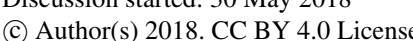

**Figure 3.** Maps of difference in simulated brightness temperatures [K] between the newer liquid water permittivity models and the current Liebe89 for 37 v brightness temperatures co-located to corresponding SSMIS-F17 observations. Means are computed in each 2.5°lat x 2.5°lon bin and over the time period 1 to 31 August 2016. White coloured areas correspond to areas where data is not assimilated, as mentioned in section 2.2.





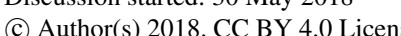

Figure 4. As Fig. 3 but for channel 92 v.





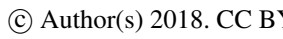

(a) Liebe93

(b) Stogryn95

(c) Ellison07

(d) Rosenkranz15

Difference in simulated brightness temperatures [K]

-0.50 -0.40 -0.30 -0.20 -0.10 -0.05 0.00 0.05 0.10 0.20 0.30 0.40 0.50

(e) TKC16

**Figure 5.** As Fig. 3 but for channel 150 h.





(a) Liebe93

(b) Stogryn95

(c) Ellison07

(d) Rosenkranz15

(e) TKC16

**Figure 6.** As Fig. 3 but for channel $183 \pm 6$ h.





**Table 3.** Mean difference in simulated brightness temperature [K] between the newer liquid water permittivity models and the current Liebe89 for 37 v, 92 v and 150 h at locations of all selected SSMIS-F17 observations over the time period 1 to 31 August 2016 for the northern hemisphere (NH: 20°N to 90°N), Tropics (20°N to 20°S) and southern hemisphere (SH: 20°S to 90°S).

| liquid water permittivity model | | 37 v | 92 v | 150 h |
|---|---|---|---|---|
| | NH | -0.009 | -0.018 | 0.003 |
| Liebe 93 | Tropics | -0.004 | -0.003 | 0.007 |
| | SH | -0.062 | -0.100 | 0.003 |
| | NH | -0.055 | -0.084 | -0.012 |
| Stogryn95 | Tropics | -0.033 | -0.041 | -0.005 |
| | SH | -0.177 | -0.266 | -0.041 |
| | NH | -0.053 | -0.050 | -0.005 |
| Ellison07 | Tropics | -0.033 | -0.042 | -0.011 |
| | SH | -0.098 | 0.008 | 0.037 |
| | NH | -0.045 | -0.054 | -0.039 |
| Rosenkranz15 | Tropics | -0.025 | -0.026 | -0.019 |
| | SH | -0.201 | -0.226 | -0.247 |
| | NH | -0.042 | -0.082 | -0.056 |
| TKC16 | Tropics | -0.022 | -0.034 | -0.020 |
| | SH | -0.195 | -0.288 | -0.271 |

the different permittivity models if the frequency increases (as mentioned in section 3.1). That means, at 92 v changes in the absorption (and, hence, brightness temperature) can be seen for warmer clouds in the subtropics. For discussion of the change in the difference between observed and simulated brightness temperatures (FG departures) see the appendix A.

### 3.3 Cold air outbreaks

5   Despite small monthly mean differences in the simulated brightness temperature (or first guess [FG]) among the six liquid water permittivity models much larger differences in simulated brightness temperature can be seen if we focus specifically on supercooled liquid water clouds. An example is the high latitudes of the southern hemisphere during austral winter which are marked by the occurrence of supercooled liquid water as illustrated for a 12-hour cycle centred around 3 UTC on 30 August 2016 in Fig. 7. Here, the differences in FG between TKC16 and Liebe89 are shown at 92 v and at $183 \pm 6$ h, with the

10  corresponding model cloud liquid water path, model snow water path and observation errors.







(a) Cloud liquid water path (kg m$^{-2}$)

(b) Snow water path (kg m$^{-2}$)

(c) Observation error (K)

(d) Difference in simulated brightness temperature (K) between TKC16 and Liebe89 at 92 v GHz.

(e) Difference in simulated brightness temperature (K) between TKC16 and Liebe89 at 183±6 GHz.

**Figure 7.** Maps for a) cloud liquid water path, b) snow water path, c) observation error with Liebe89, d) difference in FG at 92 v between TKC16 and Liebe89 and e) difference in FG at 183 ± 6 GHz between TKC16 and Liebe89 for areas of the southern mid- to high latitudes excluding land and sea ice for a 12-hour window centred at 3 UTC on 30 August 2016, co-located to SSMIS-F17 observations. Cross-hatched areas represent land and white areas have no data due to e.g. screening or quality control (see section 2.2)





The FG at 92 v simulated at SSMIS-F17 locations for TKC16 is reduced compared to Liebe89 by 0.5 K to 1.5 K (Fig. 4d). Cadeddu and Turner (2011) show in their Fig. 2 that changes in brightness temperatures of this order happen for temperatures higher than -9°C and for clouds with little liquid water of about $0.1\,\mathrm{kg\,m^{-2}}$. If temperatures were colder (T = -19°C) or if the clouds were slightly thicker (around $0.25\,\mathrm{kg\,m^{-2}}$) the change in brightness temperature would be already 2 K to 3 K. Their

finding goes along with the fact that the largest differences in simulated brightness temperatures at 92 v GHz occur in non-frontal cloud systems at higher latitudes characterized by a liquid water of about 0.2 to $0.4\,\mathrm{kg\,m^{-2}}$. In contrast, the largest changes at $183 \pm 6\,\mathrm{h}$ occur in specific conditions, e.g. at 30°S, 120°W. This could be related to cases of supercooled liquid water inside frontal systems.

The observed change in FG at 92 v of about 1 K is much smaller than the typical observation error of about 4 K to 10 K

in these regions (see Fig. 7c). Thus, using a different permittivity formulation than Liebe89 might only have a small impact on the analysis in a NWP system. The question arises if this small difference in simulated brightness temperature is too small compared to what one would expect from large differences among the permittivity models seen for supercooled liquid water clouds. In this case study, clouds with supercooled liquid water between 40°S and 60°S are usually located around 1 to 2 km height inside the forecast model, where temperatures reach between approximately 260 K to 270 K. That means,

observed changes in absorption are consistent with Fig. 2b, which shows small differences between models. However, it could be seen that clouds are located in much cooler situations south of 60°S, where microwave imager observations are currently not assimilated. Including these observations in the future, one might expect to see larger differences in simulated brightness temperature through the use of a new permittivity model.

## 4   Choice of permittivity model in RTTOV-SCATT

As shown before, most permittivity models slightly reduce the simulated brightness temperature compared to Liebe89 with two exceptions, Ellison07 and Liebe93, which both increase the simulated brightness temperature at higher frequencies in the higher latitudes of the southern hemisphere. In order to find the best choice for the RTTOV-SCATT permittivity model used inside the Integrated Forecast System (IFS) we look at different measures to quantify if the fit between model and observations is improved. Here, results are based on the monitoring experiments.

### 4.1   Different measures of fit

One measure of fit is the comparison of the standard deviation of FG departures using Liebe89 as a reference for observations from SSMIS-F17, as shown in Fig. 8. For most permittivity models the standard deviation in FG departure is reduced compared to Liebe89, as shown in Fig. 8a. The largest reduction occurs at 92 h for TKC16 of about 1.5%. This signal is more pronounced in the southern hemisphere (Fig. 8b), where TKC16 shows a reduction of about 2.6% at 92 h due to the stronger presence

of supercooled liquid water clouds during austral winter. In the southern hemisphere, Ellison07 shows a significant increase in FG departure standard deviation at 92 v GHz and at $183 \pm 6\,\mathrm{h}$, whereas Rosenkranz15 and TKC16 show an increase at $183 \pm 6\,\mathrm{h}$, only. To study the effect introduced by a changed permittivity model in more detail, focus is put on the results from





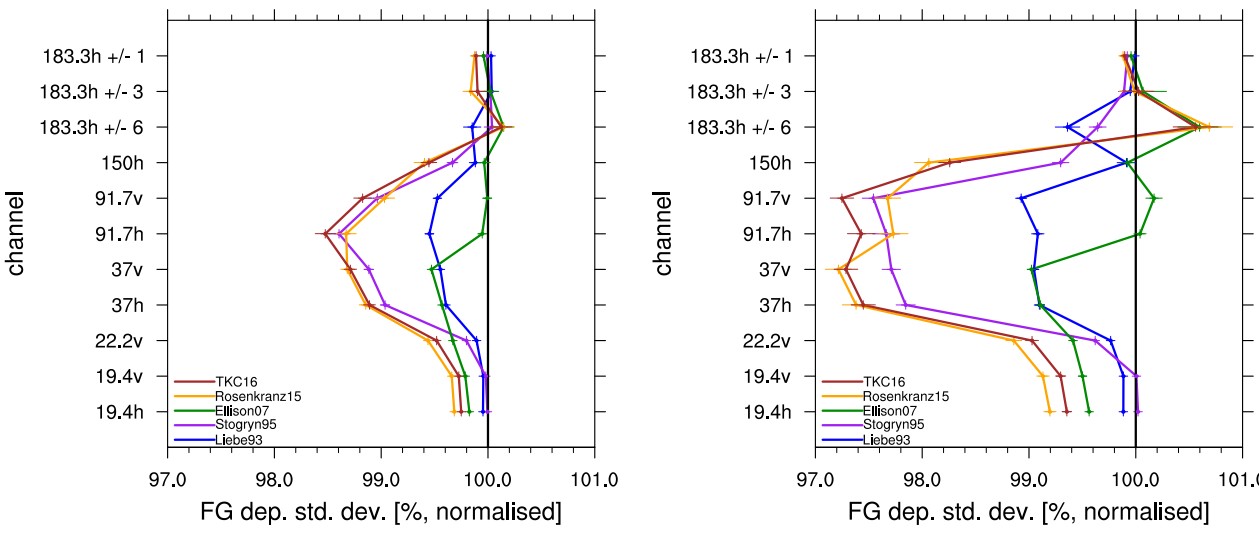

(a) All active and passive data (including areas of cold air outbreaks).

(b) All active and passive data (including areas of cold air outbreaks) in southern hemisphere (20°S - 90°S).

**Figure 8.** Standard deviation in FG departures from SSMIS-F17 for different channels normalised by the standard deviation of results from Liebe89. The horizontal bars indicate 95% confidence range. Results cover the time period from 1 to 31 August 2016. Different colours refer to different permittivity models, as shown in the figure.

the southern hemisphere (20°S - 90°S) for the remainder of this study. As discussed in section 3.3, there is a higher occurrence of supercooled liquid water clouds during austral winter in the southern hemisphere and, hence, the effects through a change in permittivity model are more pronounced. Results would be similar for other regions with supercooled liquid water.

Typically, a reduction of the standard deviation in FG departure can be interpreted as a better fit between observations and
first guess. However, for all-sky observations this measure is affected by the "double-penalty" effect. That means, better scores could be achieved if no clouds or precipitation are forecasted than by forecasting them at the wrong location or wrong time (Geer and Baordo, 2014). Additionally, compensating biases could yield a reduction in standard deviation in FG departures even if the physical realism of the absorption model is getting worse, e.g. too much scattering could be compensated by too much absorption. An alternative measure which is resistant to the double penalty effect (but unfortunately not to competing
biases) is to compare histograms of observed and simulated brightness temperatures, as done in Fig. 9 for samples of the southern hemisphere.

Here, the histograms among the different liquid water permittivity models are very similar for the different frequencies. Only slight differences are found. For example, at 92 v TKC16 and Rosenkranz15 show better fits between 215 K and 270 K with observations than Liebe89, Liebe93 and Ellison07 (Fig. 9 a, b), which is probably linked to the large reduction in FG departure
standard deviation in Fig. 8b. At 150 h slightly improved fits between 160 K and 180 K, and slightly degraded fits between 190 K and 210 K can be seen for TKC16 and Rosenkranz15 (Fig. 9 c, d). That means for low brightness temperatures at 92 v and





(a) Occurrence of brightness temperatures at 92 v.

(b) Normalised difference in occurrence at 92 v.

(c) Occurrence of brightness temperatures at 150 h.

(d) Normalised difference in occurrence at 150 h.

(e) Occurrence of brightness temperatures at 183 ± 6 h.

(f) Normalised difference in occurrence at 183 ± 6 h.

**Figure 9.** Histograms of simulated and observed brightness temperatures [K] using different liquid water permittivity models with the right panel showing the normalised difference in occurrence of simulated brightness brightness temperatures relative to observations. Bin size is 5 K. Results cover the time period from 1 to 31 August 2016 for the southern hemisphere (20°S to 90°S). Different colours refer to different permittivity models, as shown in the figure.





150 h, TKC16 and Rosenkranz15 decrease the absorption for supercooled liquid water clouds allowing more colder brightness temperatures, which agrees better with observations. Mostly small degraded fits for TKC16, Rosenkranz15 and Ellison07 are seen between 205 K and 240 K at $183 \pm 6$ h (Fig. 9 e, f), which could explain the increase in FG departure standard deviation in Fig. 8b. Very small differences in the histograms for the different permittivity models are found at $183 \pm 1$ GHz, h-polarised

(not shown). Overall, these are only small differences and the main discrepancy between the forecast model and observations is likely dominated by other effects than permittivity.

Another measure to evaluate fits between model and observations is to look at histograms of FG departures, as done in Fig. 10. It can be seen that Liebe93, Stogryn95, Rosenkranz15 and TKC16 slightly reduce the number of occurrences of large negative FG departures and increase the number of occurrences of large positive departures compared to Liebe89 at 92 v

(Fig. 10 a, b), at 150 h (Fig. 10 c, d) and at 37 v (not shown). At these channels Rosenkranz15 and TKC16 show the largest changes in numbers. This is not surprising as Rosenkranz15 and TKC16 reduce the simulated brightness temperature the most compared to the other permittivity models (see Table 3). At $183 \pm 6$ h only a small increase in the numbers at large positive FG departures can be seen for Ellison07, Rosenkranz15 and TKC16 (Fig. 10 e, f) compared to Liebe89, which probably explains the degraded fits in FG departure standard deviation (Fig. 8). This increase can be explained by low absorption values of

TKC16, Rosenkranz15 and Ellison07 compared to Liebe89 at low temperatures (supercooled liquid water), as shown in Fig. 2a. The low absorption causes smaller simulated brightness temperatures leading to an even larger difference between FG and observations (= more positive FG departures). From Fig. 6 and Fig. 7e it seems that these reduced brightness temperatures occur mostly in frontal systems in the southern hemisphere, where large FG departures are found more regularly e.g. due to displacement errors between observations and simulation.

To characterize large FG departures the skewness can be used as done by Geer and Baordo (2014). If the skewness is positive the histogram of FG departures has a large tail to the right (more large negative FG departures than large positive FG departures). The larger the skewness the more large positive FG departures exist. Rosenkranz15 and TKC16 show a larger value in skewness in FG departure (see Fig. 11) than Liebe89 at 37 v. However, their standard deviation (and mean; not shown) in FG departure is significantly smaller, as shown before in Fig. 8b. At 92 v and 150 h the skewness is smaller than Liebe89, which

goes along with a reduced standard deviation. Only at $183 \pm 6$ h skewness and standard deviation are increased compared to Liebe89. In other words, the change in skewness is associated with the change in FG departure standard deviation for frequencies higher than 37 GHz.

Nevertheless, the use of different permittivity models does not fundamentally change the shape of histograms of FG departures or brightness temperatures. If the spread amongst the permittivity models is interpreted as an indication of their likely

uncertainty levels, permittivity errors are a minor factor and do not explain the bigger picture of the differences between observations and forecast model. The degradation in FG departure standard deviation at $183 \pm 6$ h is, however, genuine and is investigated in the next section.



(a) Occurrence of FG departures at 92 v.

(b) Normalised difference in occurrence at 92 v.

(c) Occurrence of FG departures at 150 h.

(d) Normalised difference in occurrence at 150 h.

(e) Occurrence of FG departures at 183 ± 6 h.

(f) Normalised difference in occurrence at 183 ± 6 h.

**Figure 10.** Histograms of FG departures [K] using different liquid water permittivity models with the right panel showing the normalised difference for the newer permittivity models relative to Liebe89. Bin size is 5 K. Results cover the time period from 1 to 31 August 2016 for the southern hemisphere (20°S to 90°S). Different colours refer to different permittivity models, as shown in the figure.





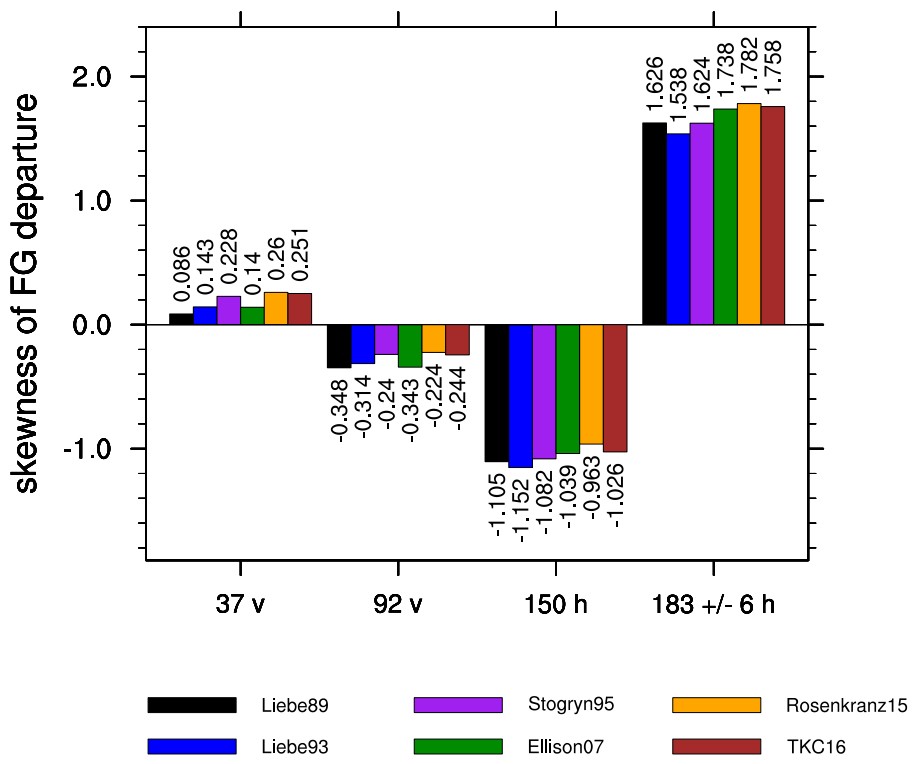

**Figure 11.** Skewness in FG departure distribution for the different liquid water permittivity models at 37 v, 92 v, 150 h and 183 ±6 h co-located to SSMIS-F17 observations over the time period 1 to 31 August 2016 for the southern hemisphere (SH: 20°S to 90°S).

## 4.2 Degradation at $183 \pm 6$ h

The degradation at $183 \pm 6$ h can be seen in a larger standard deviation in FG departure for TKC16, Rosenkranz15 and Ellison07 compared to Liebe89. The reduction in absorption and, hence, simulated brightness temperature causes larger differences compared to observations, which has mostly been associated with cases of mid-latitude frontal systems (not shown). Here, the

5 compensating effect of absorption by liquid cloud droplets and scattering by ice and snow may play a key role. Fig. 12a shows the normalised standard deviation of FG departures from SSMIS-F17 for samples with only liquid hydrometeors and a large cloud amount. Hereby, we use a symmetric measure of cloud amount C37 as defined in Geer and Bauer (2010) which is based on the polarization difference at 37 GHz and is an average of the models and the observed cloud amount. Note, that only the most intense convection shows 100% for C37. That means, the chosen value of C37 = 5% should capture scenes with enough

10 clouds, which avoids studying effects of non-cloudy condition. Here, the degradation at $183 \pm 6$ h for Ellison07, Rosenkranz15 and TKC16 reduces to the same level as the other permittivity models with most of the other improvements remaining. Even





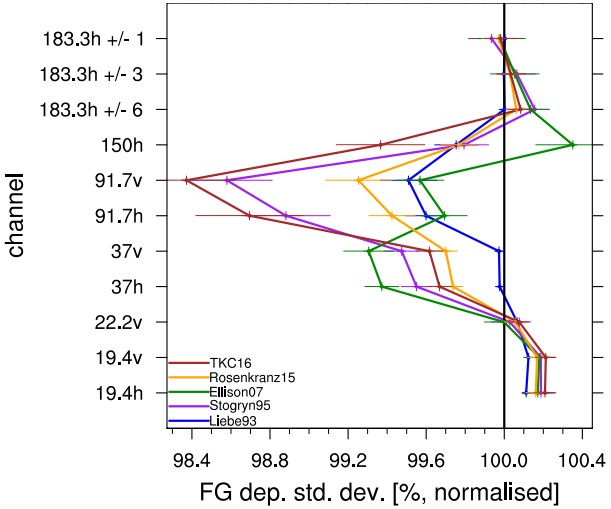

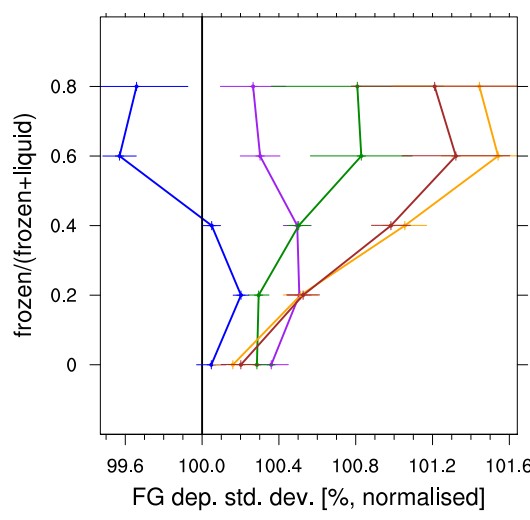

(a) Standard deviation in FG departures for samples with no frozen hydrometeors.

(b) Standard deviation in FG departures at $183 \pm 6$ GHz.

**Figure 12.** Standard deviation in FG departures from SSMIS-F17 normalised by Liebe89 for a) different channels for samples with no frozen hydrometeors containing some cloud (about 16.8% of the full sample size from the southern hemisphere) and for b) different ratios of frozen hydrometeor amount versus total hydrometeor amount at $183 \pm 6$ h. The horizontal bars indicate 95% confidence range. Results cover all active and passive data (including areas of cold air outbreaks) in the southern hemisphere ($20°$S - $90°$S) from 1 to 31 August 2016. Different colours refer to different permittivity models, as shown in the figure.

though the sample size is reduced in Fig. 12a results prove that scattering by frozen hydrometeors is related to the degradation at $183 \pm 6$ h. Fig. 12b shows how the FG departure standard deviation at $183 \pm 6$ h changes with the ratio of frozen hydrometeor amount to total hydrometeor amount for the same sample. The higher this ratio the more Ellison07, Rosenkranz15 and TKC16 become degraded, and Liebe93 and Stogryn95 become less degraded or are improved compared to Liebe89. If we plot the

change in FG departure standard deviation at $183 \pm 6$ h as a function of integrated cloud liquid water (CWP) for only liquid clouds we would see that for Rosenkranz15, TKC16 and Liebe93 the FG departure standard deviation is comparable to Liebe89 or improves with an increase in CWP but for Ellison07 and Stogryn95 the fit degrades as CWP increases (not shown). This shows that the degradation only occurs in areas with frozen hydrometeors and strong scattering.

In general, absorption increases the brightness temperature at $183 \pm 6$ GHz whereas scattering decreases for situations when

the (cold) surface is partly visible. In these cases, any biases in the representation of absorption have the potential to be compensated by biases in the representation of scattering. If scattering were already excessive at $183 \pm 6$ h, then a reduction in absorption by using TKC16, Rosenkranz15 or Ellison07 would decrease the brightness temperature even more. In other words, the compensation effect of too much absorption and too much scattering would mean that TKC16, Rosenkranz15 and Ellison07 could erroneously appear worse compared to the other permittivity models, which show higher absorption values at

183 GHz. Here, two things could cause excessive scattering. Firstly, too much frozen hydrometeors generated by the forecast



model and, secondly, too much scattering by the scattering model in e.g. frontal systems. The later case seems more likely when looking at results by Geer and Baordo (2014). They show in their Fig. 8b that the sector snowflake shape used for snow in the scattering model inside RTTOV-SCATT produces positive FG departures around $1\,\mathrm{K}$ at $183 \pm 6\,\mathrm{GHz}$ in mid- to high latitudes. That suggests, the scattering model causes too small brightness temperatures due to excessive scattering. This

excessive scattering should explain the degradations seen for Ellison07, Rosenkranz15 and TKC16 at $183 \pm 6\,\mathrm{h}$ in frontal systems at higher latitudes.

## 5   Impact on assimilation system

To properly assess the impact of the different liquid water permittivity models on the assimilation system targeted assimilation experiments are performed, as described in section 2.3. Two sets of experiments are conducted. The first set of assimilation

experiments uses the same configuration as the monitoring experiments, which uses supplementary observations containing cold-air outbreak areas and low water vapour areas, and allows the generation of additional supercooled liquid water clouds inside the forecast model; **plusSLW**. The second set of experiments simply uses the default setup, which does not use observations containing cold-air outbreak areas and low water vapour areas; **screen**. In order to assess the impact the forecast scores and fits to observations have been analysed. Results are only shown for Stogryn95 and Rosenkranz15, because Liebe93 and

Ellison07 have been identified to show the smallest improvements (see section 4), and TKC16 is very similar to Rosenkranz15.

It is found that using different formulations of permittivity shows a neutral impact on forecast scores in terms of a change in root-mean-square error in humidity, temperature and wind in the long- and short-term for **plusSLW** and **screen** (not shown). This is likely related to the fact that the introduced change in simulated brightness temperatures is small both relative to observation error (e.g. Fig. 7c) and relative to differences between observations and forecast model (Fig. 9, Fig. 10). However,

fits of the first guess forecast to humidity sensitive observations are altered through a change of the liquid water permittivity model. For example, Rosenkranz15 and Stogryn95 improve fits to the humidity sensitive channels of the Advanced Technology Microwave Sounder (ATMS; channels 18 -22) in the southern hemisphere for **plusSLW** compared to Liebe89 (Fig. 13a). An improvement seen for ATMS is a result of an improved FG field in humidity because ATMS is only assimilated in clear-sky conditions (section 2.2) and, hence, cannot be affected directly by a change in liquid water permittivity model inside the

RTTOV-SCATT.

This is different to the Microwave Humidity Sounding (MHS) instrument, which is assimilated under all-sky conditions. Here, using Rosenkranz15 degrades the fit to channel 5 ($183 \pm 7\,\mathrm{GHz}$, v-polarised), whereas Stogryn95 improves it to a similar extent in **plusSLW** and in **screen** (Fig. 13b and Fig. 13d, respectively). The degradation for Rosenkranz15 is most likely caused by the excess scattering in mid-latitude frontal systems which is not as much compensated by an excess in absorption as in

Liebe89 (discussed in section 4). A similar change is found for most $183\,\mathrm{GHz}$ channels of the Sondeur Atmospherique du Profil d'Humidite Intertropicale par Radiometrie (SAPHIR) for both permittivity models in **plusSLW** (not shown).

As expected, improved fits to microwave imagers are found, e.g. in fits to SSMIS (Fig. 13c), similar for the GPM Microwave Imager (GMI) and the Advanced Microwave Scanning Radiometer 2 (AMSR2; not shown). Here, Rosenkranz15 shows larger





improvements than Stogryn95, even for **screen**. Interestingly, only when cold-air outbreak areas and low water vapour areas are included (**plusSLW**) a degradation is found at $183 \pm 6\,$h (channel 9) for Rosenkranz15. A likely explanation could be that the screening also removes some of those mid-latitude frontal areas with the moderate brightness temperature changes (as seen in Fig. 7e), not just cold air outbreaks. The reason being is that 80% of cold-air outbreaks occur in association with a cyclonic

flow (Papritz et al., 2015).

Additionally, mean changes in the bias of FG departures at 37 v and at 92 v have been analysed for microwave imagers in the southern hemisphere (not shown). For SSMIS the bias changed by about 0.2 K which led to a reduction in bias in **plusSLW** to 0 K and slight increase to 0.3 K for **screen**. For GMI and AMSR2 the bias between -0.25 K to -0.5 K has been reduced by about 0.2 K for both **screen** and **plusSLW**. Fits to temperature sensitive observations (e.g. the Advanced Microwave Sounding

Unit - A, AMSU-A) and wind (e.g. atmospheric motion vectors) are neutrally affected by the different choices of permittivity models for **screen** and **plusSLW** (not shown).

## 6 Conclusions

We have studied the effect of six different permittivity formulations on simulated brightness temperatures (first guess, FG) and the impact on the assimilation system using the integrated forecast system (IFS). As shown already by e.g. Kneifel et al.

(2014), newer liquid water permittivity models are known to give significant lower values of absorption for supercooled liquid water at microwave frequencies above 19 GHz.

A model configuration is used which allows the generation of more and colder supercooled liquid water than available in earlier IFS versions. Firstly, the limit of the existence of supercooled liquid water has been changed from -23°C to -38°C for convective mixed phase clouds and, secondly, the model physics upgrade in IFS cycle 45R1 allowed the generation of

purely supercooled liquid water inside surface driven shallow clouds. This change was motivated by findings showing a lack of supercooled liquid water in cold-air outbreak regions inside the forecast model (Forbes et al., 2016). Even though this configuration misses the generation of some supercooled liquid water in congestus clouds and deep convection clouds, it seems good enough to study the impact of different liquid water permittivity models including clouds having supercooled liquid water. Additionally, microwave imager observations in these regions have been included in the assimilation, which are usually

screened due to a systematic model bias.

Most of the permittivity formulations reduce the simulated brightness temperatures slightly compared to Liebe89 due to their smaller values in absorption. Especially, in areas with supercooled liquid water (e.g. cold-air outbreaks) the largest reduction in simulated brightness temperatures could be observed. There are just two exceptions: Liebe93 and Ellison07. Due to slightly larger values in absorption for higher microwave frequencies Liebe93 and Ellison07 increase the simulated brightness

temperature in areas of supercooled liquid water. The newer permittivity formulations by Rosenkranz15 and TKC16 show the largest reductions together with Stogryn95. Using TKC16 reduces the simulated brightness temperature about 0.5 K to 1.5 K at 92 v for regions with supercooled liquid water. Using a forecast model allowing the generation of purely supercooled liquid water congestus clouds or deep convection might be able to reduce the brightness temperatures even further. However, this



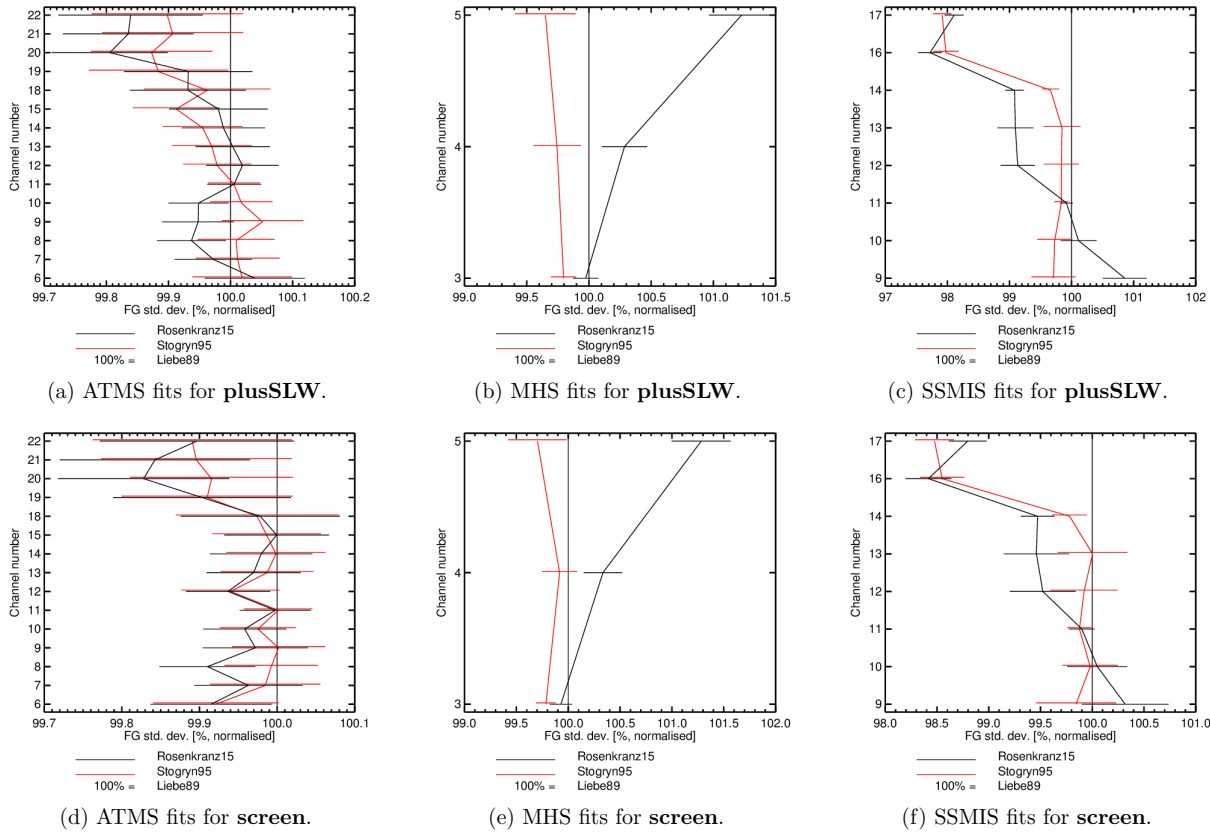

(a) ATMS fits for **plusSLW**.  (b) MHS fits for **plusSLW**.  (c) SSMIS fits for **plusSLW**.

(d) ATMS fits for **screen**.  (e) MHS fits for **screen**.  (f) SSMIS fits for **screen**.

**Figure 13.** Standard deviation in FG departures in the southern hemisphere of ATMS, MHS and SSMIS for Rosenkranz15 and Stogryn95 normalised by Liebe89 for **plusSLW** and for **screen**. Different colours refer to different liquid water permittivity models, as shown in the figure. The horizontal bars indicate 95% confidence range. Results cover the time period from 1 June to 30 September 2016.

cannot be concluded from the set of experiments presented in this study and more targeted studies are necessary to confirm this hypothesis.

On a global scale, the differences between the permittivity models are small and cannot explain the main discrepancy between model and observations. However, the biggest improvements in terms of observational fits to microwave imagers
5 could be seen for the new permittivity models TKC16 and Rosenkranz15 for frequencies below 183 GHz. Some degradation at $183 \pm 6$ GHz from SSMIS and MHS have been seen for Ellison07, Rosenkranz15 and TKC16. This degradation seems to occur in clouds containing some supercooled liquid water in mid-latitude frontal systems. Here, the compensating biases in the scattering model and in the absorption model play most likely a major part. Geer and Baordo (2014) have shown, that the current choice of sector snow flake shape and the choice in particle size distribution in the scattering model inside RTTOV-SCATT
10 introduces excessive scattering in the higher latitudes. This excessive scattering seems to be less compensated through liquid



water absorption when using Ellison07, Rosenkranz15 or TKC16. To address this apparent degradation studies are planned to re-examine how the forecast model represents clouds and precipitation, how the data assimilation framework handles cloud- and precipitating affected observations, how we can improve the construction of the observation operator and how observation errors are treated in the all-sky assimilation at ECMWF (Geer et al., 2017a).

To properly test the impact of the different permittivity models on the assimilation system targeted assimilation experiments have been conducted. It could be shown that the forecast is only neutrally affected by a change in permittivity model, which is probably due to the large observation errors relative to changes in brightness temperatures caused by different liquid water permittivity models. Nevertheless, improved fits to independent observations, like the humidity channels of ATMS are found for the southern hemisphere. In the future when forecast models will be capable to generate enough supercooled liquid water

clouds and the assimilation system will use microwave observations in these regions the impact of the permittivity formulation will be even more crucial. But already now most of the observational fits to humidity and cloud sensitive observations are improved and forecast scores are not degraded by using the liquid water permittivity formulation by Rosenkranz (2015) or Turner et al. (2016).

In light of those results: i) small impact on simulated brightness temperatures in regions with a relatively large systematic

error, ii) neutral impact on forecast scores and iii) difficulty in balancing good and bad changes because of the compensating biases in scattering and absorption, one has to ask the question if such a NWP closure study is actually able to find the "best" liquid water permittivity model? We would argue that it is possible, at least to reject the worst models. Such a closure study has the unique ability to quantify the global effect of supercooled liquid water permittivity changes in a high-quality model atmosphere, and not just locally as done through comparisons with observations from ground or under idealized conditions

in laboratory experiments. Additionally, it is found that using different liquid water permittivity models shows a sensitivity to independent data sets (e.g. for ATMS). Lastly, it is reassuring that the newest permittivity models Rosenkranz15 and TKC16, which are based amongst other things on the most up-to-date observations do also give the best fits to the microwave imagers SSMIS, GMI and AMSR2. Our results indicate that either TKC16 or Rosenkranz15 should be used inside RTTOV-SCATT; both showing a similar level of improvement. For now that would encompass microwave frequencies which are less prone

to compensating biases in the scattering and absorption model, i.e. below 183 GHz. Looking into the future, where we want to assimilate microwave frequencies up to 1 THz, we favour the use of the Rosenkranz15 permittivity model inside RTTOV-SCATT, as it has also been constructed for higher microwave frequencies, whereas TKC16 is only valid up to 500 GHz.

## Appendix A: Change in FG departures

Fig. A1, Fig. A2, Fig. A3 and Fig. A4 show the geographical distribution of mean differences between the different permittivity

models in observed minus simulated brightness temperatures (FG departures) compared to Liebe89 at 37 v, 92 v, 150 h and $183 \pm 6$ h respectively. The largest changes in FG departures can be seen for Stogryn95, Rosenkranz15 and TKC16 in the southern mid- to high latitudes for frequencies up to 150 GHz. Hereby, the absolute value in FG departure is reduced by about 0.3 K and 0.6 K at 37 v and at 92 v, respectively, with an additional increase in FG departures of about 0.3 K in the northern





mid- to high latitudes at 92 v. At 150 h a slight increase in FG departures is shown for the southern mid-latitudes and a decrease is shown for the southern higher latitudes of about 0.6 K for Rosenkranz15 and TKC16, only. No large changes can be seen for $183 \pm 6$ h. The mean changes in FG departures are plotted for the monitoring experiments which include SSMIS-F17 observations from areas, which are usually screened in the default setup as described in Sec. 2.3.3. That means, in these plots

5    only changes in FG departure due to a change of the observation operator are highlighted.





(a) Liebe93

(b) Stogryn95

(c) Ellison07

(d) Rosenkranz15

(e) TKC16

**Figure A1.** Maps of difference in observed minus simulated brightness temperatures [K] between the newer liquid water permittivity models and the current Liebe89 for 37 v brightness temperatures co-located to corresponding SSMIS-F17 observations. Means are computed in each 2.5°lat x 2.5°lon bin and over the time period 1 to 31 August 2016. White coloured areas correspond to areas where data is not assimilated, as mentioned in section 2.2.







(a) Liebe93

(b) Stogryn95

(c) Ellison07

(d) Rosenkranz15

(e) TKC16

**Figure A2.** As Fig. A1 but for channel 92 v.



(a) Liebe93

(b) Stogryn95

(c) Ellison07

(d) Rosenkranz15

(e) TKC16

**Figure A3.** As Fig. A1 but for channel 150 h.





(a) Liebe93

(b) Stogryn95

(c) Ellison07

(d) Rosenkranz15

Difference in abs(mean) FGdep [K]

-1.00  -0.83  -0.67  -0.50  -0.33  -0.17  0.00  0.17  0.33  0.50  0.67  0.83  1.00

(e) TKC16

**Figure A4.** As Fig. A1 but for channel $183 \pm 6$ h.





*Competing interests.* No competing interests are present.

*Acknowledgements.* Katrin Lonitz work at ECMWF is funded by the EUMETSAT fellowship programme. Stefan Kneifel is thanked for making his code for the Liebe93, Stogryn95 and Ellison07 permittivity model available. Robin Hogan is acknowledged for encouraging the authors to undertake this study.




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
