# Peer review of "Assessing the impact of different liquid water permittivity models on the fit between model and observations"

_Atmospheric Measurement Techniques, 2018_

## Referee Comment (RC1) · Anonymous Referee #1 · 13 Jun 2018

General comments: The manuscript reports on tests of several models for liquid-water dielectric constant within the Integrated Forecast System of ECMWF. The models are compared with respect to calculated microwave brightness temperatures and differences from satellite observations. Weather forecasting is an important application of a dielectric model, and this manuscript makes a substantial contribution on the subject of modeling the electrical properties of clouds. This question is within the scope of AMT. The manuscript recognizes difficulties such as the treatment of scattering within the radiative-transfer algorithm, but shows that some meaningful conclusions can be drawn about the dielectric models.

[Figure]

Specific comments:

1. The conclusions section provides a good summary of the paper. However, the abstract's brief statement of the recommendation may mislead, since Rosenkranz15, TKC16, and Stogryn95 yielded nearly equivalent improvements.

2. The closure experiments described in this paper are based on ECMWF-IFS, and it should be said that other forecast systems could conceivably yield different results.

3. In the second paragraph of Section 5 (on p.23), the reasoning is hard to follow. Lines 16-17 state that there is a neutral impact on humidity, temperature and wind. But line 23 says "An improvement seen for ATMS is a result of an improved FG field in humidity..."

Technical corrections:

4. page 3, lines 21-23. "Permittivity ... consists of a real (scattering) component, and an imaginary (absorption) component." Actually, both real and imaginary parts of permittivity are involved in scattering and absorption by spherical droplets.

5. page 7, lines 16-17. The sentence "The two most recent permittivity models Rosenkranz15 and TKC16 give about 50% of the absorption compared to Liebe89 for frequencies around the 183 GHz water vapour absorption line." belongs under the discussion of Fig.2a for 240K, starting at line 20.

---

## Referee Comment (RC2) · Anonymous Referee #2 · 17 Jun 2018

Comment to manuscript AMT-2018-152: Assessing the impact of different liquid water permittivity models on the fit between model and observations by K. Loniz and A. J. Geer.

General comments: The paper summarizes results of testing six water dielectric models in NWP models especially evaluating the performance in the regions where supercooled water ($< 0^{\circ}$C) occurs frequently. Different experiments and metrics are developed and tested to find which dielectric model yields a better fit with satellite measurements assimilated in NWP models. The results seem to point to the latest Rosenkranz, TKC, and perhaps Stogryn (for some frequencies) as the models that may agree better with observations of supercooled clouds. These type of testing is necessary as it is useful to know which model to use especially in view of the large radiative impact of supercooled clouds at high latitudes, therefore I recommend publication. The paper is generally well written and organized. I have minor comments and few questions for the authors.

1) The conclusions in the abstract not entirely consistent with conclusions in the discussion section.

2) Page 4 line 15: *"For SSMIS-F17 an observation error of 1.8 K is used in clear-sky conditions (C37<0.02), which increases linear up to 18 K for very cloudy situations with C37>0.42. The higher the observation error the less impact the observation has on the analysis. More details can be found in Geer and Bauer (2011)."*
Are these observation errors theoretical or are they based on actual observations?

3) Fig.2 is interesting as it shows discrepancies between models at higher frequencies even in non-supercooled liquid. In view of the new ICI satellite that will use frequencies > 200 GHz this will require some additional validation. Perhaps in Table 3 it may be worth adding a few frequencies in the sub-mm range, perhaps 325, 448, and 664 GHz?

4) In Fig. 7 is the cloud liquid water path estimated from microwave observations using the L89 model?

5) I am not really sure how to interpret fig 9 since the bin size (5 K) is much bigger that the differences in brightness temperatures between the models. Not sure this figure adds much to the discussion. Fig. 10 seems to provide more information.

6) Page 19 line 23 *"Rosenkranz15 and TKC16 show a larger value in skewness in FG departure (see Fig. 11) than Liebe89 at 37 v."*
Stogryn95 model seems very similar.

7) Page 23 lines 16-25. This sentence appears confused at least isn't clear to me what the

authors are trying to say. First they say there is a neutral impact in the rms error in humidity, but then they say that there is an improvement in ATMS data, but then they say ATMS data are not affected by permittivity models.

---

## Author Comment (AC1) · 23 Nov 2018

**Anonymous Referee No.1**

**Specific comments:**

*1. The conclusions section provides a good summary of the paper. However, the abstract's brief statement of the recommendation may mislead, since Rosenkranz15, TKC16, and Stogryn95 yielded nearly equivalent improvements.*

We agree with the referees comment and amended the manuscript accordingly. Instead of recommending Rosenkranz15 we reworded the abstract to say that we favour Rosenkranz15 in this study because of the reasons named in the last paragraph of the conclusions. p.1 l.9-16

*2. The closure experiments described in this paper are based on ECMWF-IFS, and it should be said that other forecast systems could conceivably yield different results.*

That is true. The study undertaken is IFS specific. We edited the manuscript to highlight this fact to the reader: p.6 l.10-12

*3. In the second paragraph of Section 5 (on p.23), the reasoning is hard to follow. Lines 16-17 state that there is a neutral impact on humidity, temperature and wind. But line 23 says "An improvement seen for ATMS is a result of an improved FG field in humidity…"*

The statement in lines 16/17 on page 23 refers to the impact on forecast scores in terms of a change in root-mean-square error in humidity, temperature and wind in the long- and short-term, which was neutral. However, analysis-based verification can be unreliable at short ranges due to correlations between the forecast and the reference, so we would place more reliance on verification against observations here, which is done using the humidity sensitive instrument ATMS (and others). p.22 l.20-22

**Technical corrections:**

*4. page 3, lines 21-23. "Permittivity ... consists of a real (scattering) component, and an imaginary (absorption) component." Actually, both real and imaginary parts of permittivity are involved in scattering and absorption by spherical droplets.*

We edited the sentence to: "Permittivity ... consists of a real component, and an imaginary component." p3 l.24/25

*5. page 7, lines 16-17. The sentence "The two most recent permittivity models Rosenkranz15 and TKC16 give about 50% of the absorption compared to Liebe89 for frequencies around the 183 GHz water vapour absorption line." belongs under the discussion of Fig.2a for 240K, starting at line 20.*

Done. p.7 l.13-15

---

## Author Comment (AC2) · 23 Nov 2018

**Anonymous Referee No.2**

**General comments:**

The paper summarizes results of testing six water dielectric models in NWP models especially evaluating the performance in the regions where supercooled water (< 0 o C) occurs frequently. Different experiments and metrics are developed and tested to find which dielectric model yields a better fit with satellite measurements assimilated in NWP models. The results seem to point to the latest Rosenkranz, TKC, and perhaps Stogryn (for some frequencies) as the models that may agree better with observations of supercooled clouds. These type of testing is necessary as it is useful to know which model to use especially in view of the large radiative impact of supercooled clouds at high latitudes, therefore I recommend publication. The paper is generally well written and organized. I have minor comments and few questions for the authors.

1) The conclusions in the abstract not entirely consistent with conclusions in the discussion section.

Thanks for the comment. We think this is similar to comment 1 from referee 1. We edited the abstract to highlight that we favour the use of Rosenkranz15 as argued in the manuscript and to be consistent with the conclusions. p.1 l.9-16

2) Page 4 line 15: "For SSMIS-F17 an observation error of 1.8 K is used in clear-sky conditions (C37<0.02), which increases linear up to 18 K for very cloudy situations with C37>0.42. The higher the observation error the less impact the observation has on the analysis. More details can be found in Geer and Bauer (2011)." Are these observation errors theoretical or are they based on actual observations?

The observation errors are based on standard deviation in FG departure plots as a function of C37, which is described in more detail in Geer and Bauer (2011) "Observation errors in all-sky data assimilation", e.g. Fig. 8.

3) Fig.2 is interesting as it shows discrepancies between models at higher frequencies even in nonsupercooled liquid. In view of the new ICI satellite that will use frequencies > 200 GHz this will require some additional validation. Perhaps in Table 3 it may be worth adding a few frequencies in the sub-mm range, perhaps 325, 448, and 664 GHz?

Interesting point, but not yet doable. Table3 shows results for some selected frequencies of the SSMIS-F17 instrument, which does not have the requested frequencies in the sub-mm range as the ICI satellite would have. To get results as shown in Tab.3 for ICI, one has to add the ICI instrument to our configuration, which would require several months of testing and is, indeed, a new study.

4) In Fig. 7 is the cloud liquid water path estimated from microwave observations using the L89 model?

No, it's taken from the model. The caption of Fig. 7 is amended to highlight this fact. p.16

5) I am not really sure how to interpret fig 9 since the bin size (5K) is much bigger that the differences in brightness temperatures between the models.Not sure this figure adds much to the discussion. Fig.10 seems to provide more information.

After much thought, we have agreed with the referee and decided to leave out Fig.9. The discussion of Fig.9 is deleted and the remaining manuscript edited, accordingly.

6) Page 19 line 23 "Rosenkranz15 and TKC16 show a larger value in skewness in FG departure (see Fig. 11) than Liebe89 at 37 v." Stogryn95 model seems very similar.

True. However, Rosenkranz15 and TKC15 show the largest values in skeweness at 37v. We amended the sentence to reflect that. p.18 l.18/19 Furthermore, we edited the abstract and conclusion to state that Stogryn95 is indeed quite similar to Rosenkranz15 and TKC16.

7) Page 23 lines 16-25. This sentence appears confused at least isn't clear to me what the authors are trying to say. First they say there is a neutral impact in the rms error in humidity, but then they say that there is an improvement in ATMS data, but then they say ATMS data are not affected by permittivity models.

A similar statement has been made by referee 1 (#3). Hence, we would refer referee 2 kindly to read our statement to comment 3 of referee 1. However, this confusion highlighted a need to add some extra information on what FG is in our IFS-setup, which is now done and why it is important to look at observation based statistics. (p.22 l.20-22) Also, the reason why ATMS data are not affected by permittivity models has been stated on page 22, lines 18-20. Just to clarify, the liquid water permittivity model is part of the all-sky observation operator (RTTOV-SCATT) and hence, clear-sky observations do not make use of this observation operator. That means improved fits to ATMS data must be due to an improved FG field, rather an improved observation operator.